# Approximate Gaussian process inference for the drift of stochastic differential equations

**Andreas Ruttor**
Computer Science, TU Berlin
andreas.ruttor@tu-berlin.de

**Philipp Batz**
Computer Science, TU Berlin
philipp.batz@tu-berlin.de

**Manfred Opper**
Computer Science, TU Berlin
manfred.opper@tu-berlin.de

## Abstract

We introduce a nonparametric approach for estimating drift functions in systems of stochastic differential equations from sparse observations of the state vector. Using a Gaussian process prior over the drift as a function of the state vector, we develop an approximate EM algorithm to deal with the unobserved, latent dynamics between observations. The posterior over states is approximated by a piecewise linearized process of the Ornstein-Uhlenbeck type and the MAP estimation of the drift is facilitated by a sparse Gaussian process regression.

## 1 Introduction

Gaussian process (GP) inference methods have been successfully applied to models for dynamical systems, see e.g. [1–3]. Usually, these studies have dealt with discrete time dynamics, where one uses a GP prior for modeling transition function and the measurement function of the system. On the other hand, many dynamical systems in the physical world evolve in continuous time and the noisy dynamics is described naturally in terms of stochastic differential equations (SDE). SDEs have also attracted considerable interest in the NIPS community in recent years [4–7]. So far most inference approaches have dealt with the posterior prediction of state variables between observations (smoothing) and the estimation of parameters contained in the drift function, which governs the deterministic part of the microscopic time evolution. Since the drift is usually a nonlinear function of the state vector, a *nonparametric* estimation using Gaussian process priors would be a natural choice, when a large number of data is available. A recent result by [8, 9] presented an important step in this direction. The authors have shown that GPs are a conjugate family to SDE likelihoods. In fact, if an entire path of dense observations of the state dynamics is observed, the posterior process over the drift is exactly a GP. Unfortunately, this simplicity is lost, when observations are not dense, but separated by larger time intervals. In [8] this sparse, incomplete observation case has been treated by a Gibbs sampler, which alternates between sampling complete state paths of the SDE and creating GP samples for the drift. A nontrivial problem is the sampling from SDE paths *conditioned* on observations. Second, the densely sampled hidden paths are equivalent to a large number of imputed observations, for which the matrix inversions required by the GP posterior predictions can become computationally costly. It was shown in [8] that in the univariate case for GP priors with precision operators (the inverses of covariance kernels) which are differential operators efficient predictions can be realized in terms of the solutions of differential equations.

In this paper, we develop an alternative approximate *expectation maximization* (EM) method for inference from sparse observations, which is faster than the sampling approach and can also be applied to arbitrary kernels and multivariate SDEs. In the E-Step we approximate expectations over

state paths by those of a locally fitted Ornstein-Uhlenbeck model. The M-step for computing the maximum posterior GP prediction of the drift depends on a continuum of function values and is thus approximated by a sparse GP.

The paper is organized as follows. Section 2 introduces stochastic differential equations and section 3 discusses GP based inference for completely observed paths. In section 4 our approximate EM algorithm is derived and its performance is demonstrated on a variety of SDEs in section 6. Section 7 presents a discussion.

## 2 Stochastic differential equations

We consider continuous-time univariate Markov processes of the diffusion type, where the dynamics of a $d$-dimensional state vector $X_t \in \mathcal{R}^d$ is given by the stochastic differential equation (SDE)

$$dX_t = f(X_t)dt + D^{1/2}dW. \tag{1}$$

The vector function $f(x) = (f^1(x), \ldots, f^d(x))$ defines the deterministic drift and $W$ is a Wiener process, which models additive white noise. $D$ is the diffusion matrix, which we assume to be independent of $x$. We will not attempt a rigorous treatment of probability measures over continuous time paths here, but will mostly assume for our derivations that the process can be approximated with a discrete time process $X_t$ in the Euler-Maruyama discretization [10], where the times $t \in G$ are on a regular grid $G = \{0, \Delta t, 2\Delta t, \dots \}$ and where $\Delta t$ is some small microscopic time. The discretized process is given by

$$X_{t+\Delta t} - X_t = f(X_t)\Delta t + D^{1/2}\sqrt{\Delta t}\,\epsilon_t, \tag{2}$$

where $\epsilon_t \sim \mathcal{N}(0, I)$ is a sequence of i.i.d. Gaussian noise vectors. We will usually take the limit $\Delta t \to 0$ only in expressions where (Riemann) sums are over nonrandom quantities, i.e. where expectations over paths have been carried out and can be replaced by ordinary integrals.

## 3 Bayesian Inference for dense observations

Suppose we observe a path of $n$ $d$-dimensional observations $X_{0:T} = (X_t)_{t\in G}$ over the time interval $[0, T]$. Since for $\Delta t \to 0$, the transition probabilities of the process are Gaussian,

$$p_f(X_{0:T}|\mathbf{f}) \propto \exp\left[ -\frac{1}{2\Delta t}\sum_t ||X_{t+\Delta t} - X_t - f(X_t)\Delta t||^2 \right], \tag{3}$$

the probability density for the path with a given drift function $\mathbf{f} \doteq (f(X_t))_{t\in G}$ at these observations can be written as the product

$$p_f(X_{0:T}|\mathbf{f}) = p_0(X_{0:T})L(X_{0:T}|\mathbf{f}), \tag{4}$$

where

$$p_0(X_{0:T}) \propto \exp\left[ -\frac{1}{2\Delta t}\sum_t ||X_{t+\Delta t} - X_t||^2 \right] \tag{5}$$

is the measure over paths without drift, i.e. a discretized version of the *Wiener measure*, and a term which we will call *likelihood* in the following,

$$L(X_{0:T}|\mathbf{f}) = \exp\left[ -\frac{1}{2}\sum_t ||f(X_t)||^2\,\Delta t + \sum_t \langle f(X_t), X_{t+\Delta t} - X_t \rangle \right]. \tag{6}$$

Here we have introduced the inner product $\langle u, v \rangle \doteq u^\top D^{-1}v$ and the corresponding squared norm $||u||^2 \doteq u^\top D^{-1}u$ to avoid cluttered notation.

To attempt a nonparametric Bayesian estimate of the drift function $f(x)$, we note that the exponent in (6) contains the drift $\mathbf{f}$ at most quadratically. Hence it becomes clear that a conjugate prior to the drift for this model is given by a Gaussian process, i.e. we assume that for each component $\mathbf{f} \sim P_0(\mathbf{f}) = \mathrm{GP}(0, K)$, where $K$ is a kernel [11], a fact which was recently observed in [8]. We denote probabilities over the drift $\mathbf{f}$ by upper case symbols in order to avoid confusion with path probabilities. Although a more general model is possible, we will restrict ourselves to the case where the

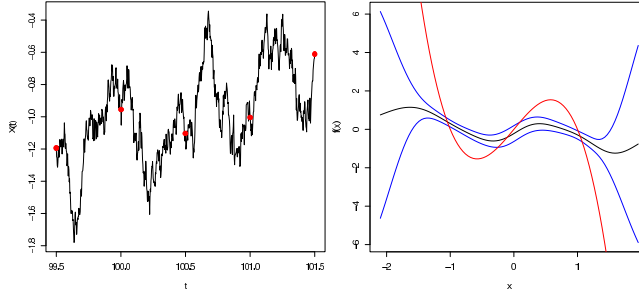

Figure 1: The left figure shows a snippet of the double well sample path in black and observations as red dots. The right picture displays the estimated drift function for the double well model after initialization, where the red line denotes the true drift function and the black line the mean function with corresponding 95%-confidence bounds (twice the standard deviation) in blue. One can clearly see that the larger distance between the consecutive points leads to a wrong prediction.

GP priors over the components $f^j(x)$, $j = 1, \ldots, d$ of the drift are independent (with usually different kernels) and we assume that we have a diagonal diffusion matrix $D = \text{diag}(\sigma_1^2, \ldots, \sigma_d^2)$. In this case, the GP posteriors of $f^j(x)$ are independent, too, and we can estimate drift components independently by ordinary GP regression. We define *data vectors* by $\mathbf{d}^j = ((X_{t+\Delta t}^j - X_t^j)/\Delta t)_{t \in G \setminus \{T\}}^\top$, the *kernel matrix* $\mathbf{K}^j = (K^j(X_s, X_t))_{s,t \in G}$, and the *test vector* $\mathbf{k}^j(x) = (K^j(x, X_t))_{t \in G}^\top$. Then a standard calculation [11] shows that the posterior process over drift functions $f$ has a posterior mean and a GP posterior variance at an arbitrary point $x$ is given by

$$\bar{f}^j(x) = \mathbf{k}^j(x)^\top \left( \mathbf{K}^j + \frac{\sigma_j^2}{\Delta t}\mathbf{I} \right)^{-1} \mathbf{d}^j, \qquad \sigma_{f^j}^2(x) = K^j(x, x) - \mathbf{k}^j(x)^\top \left( \mathbf{K}^j + \frac{\sigma_j^2}{\Delta t}\mathbf{I} \right)^{-1} \mathbf{k}^j(x). \quad (7)$$

Note that $\sigma_j^2/\Delta t$ plays the role of the variance of the *observation noise* in the standard regression case. In practice, the number of observations can be quite large for a fine time discretization, and a fast computation of (7) could become infeasible. A possible way out of this problem—as suggested by [8]—could be a restriction to kernels for which the inverse kernel, the precision operator, is a differential operator. A well known machine learning approach, which is based on a sparse Gaussian process approximation, applies to arbitrary kernels and generalizes easily to multivariate SDE. We have resorted specifically to the optimal Kullback-Leibler sparsity [1, 12], where the likelihood term of a GP model is replaced by another effective likelihood, which depends only on a smaller set of variables $\mathbf{f}_s$.

## 4 MAP Inference for sparse observations

The simple GP regression approach outlined in the previous section cannot be applied when observations are sparse in time. In this setting, we assume that $n$ observations $y_k \doteq X_{\tau_k}$, $k = 1, \ldots, n$ are obtained at (for simplicity) regular intervals $\tau_k = k\tau$, where $\tau \gg \Delta t$ is much larger than the microscopic time scale. In this case, a discretization in (6), where the sum over the microscopic grid $t \in G$ would be replaced by a sum over *macroscopic* times $\tau_k$ and $\Delta t$ by $\tau$, would correspond to a discrete time dynamical model of the form (1) again replacing $\Delta t$ by $\tau$. But this discretization would give a bad approximation to the true SDE dynamics. The estimator of the drift would give some (approximate) estimation of the mean of the transition kernel over macroscopic times $\tau$. However, this does usually not give a good approximation for the original drift. This can be seen in figure 1, where the red line corresponds to the true drift (of the so called double-well model [4]) and the black line to its prediction based on observations with $\tau = 0.2$ and the naive estimation method.

To deal with this problem, we treat the process $X_t$ for times $t$ between consecutive observations $k\tau < t < (k+1)\tau$ as a latent unobserved random variable with a posterior path measure given by

$$p(X_{0:T}|\mathbf{y}, \mathbf{f}) \propto p(X_{0:T}|\mathbf{f}) \prod_{k=1}^n \delta(y_k - X_{k\tau}), \quad (8)$$

where $\mathbf{y}$ is the collection of observations $y_k$ and $\delta(\cdot)$ denotes the Dirac-distribution encoding the fact that the process is known perfectly at times $\tau_k$. Our goal is to use an EM algorithm to compute the *maximum posterior* (MAP) prediction for the drift function $f(x)$. Unfortunately, exact posterior expectations are intractable and one needs to work with suitable approximations.

## 4.1 Approximate EM algorithm

The EM algorithm cycles between two steps

1. In the E-step, we compute the expected negative logarithm of the complete data likelihood
$$\mathcal{L}(\mathbf{f}, q) = -E_q \left[ \ln L(X_{0:T}|\mathbf{f}) \right], \tag{9}$$
    where $q$ denotes a measure over paths which approximates the intractable posterior $p(X_{0:T}|\mathbf{y}, \mathbf{f}_{old})$ for the previous estimate $\mathbf{f}_{old}$ of the drift.

2. In the M-Step, we recompute the drift function as
$$\mathbf{f}_{new} = \arg \min_{\mathbf{f}} \left( \mathcal{L}(\mathbf{f}, q) - \ln P_0(\mathbf{f}) \right). \tag{10}$$

To compute the expectation in the E-step, we use (6) and take the limit $\Delta t \to 0$ at the end, when expectations have been computed. As $f(x)$ is a time-independent function, this yields

$$
\begin{aligned}
-E_q[\ln L(X_{0:T}|\mathbf{f})] &= \lim_{\Delta t \to 0} \frac{1}{2} \sum_t E_q \left[ ||f(X_t)||^2 \Delta t - 2 \langle f(X_t), X_{t+\Delta t} - X_t \rangle \right] \\
&= \frac{1}{2} \int_0^T E_q \left[ ||f(X_t)||^2 - 2 \langle f(X_t), g_t(X_t) \rangle \right] dt \\
&= \frac{1}{2} \int ||f(x)||^2 A(x) dx - \int \langle f(x), y(x) \rangle \, dx. \tag{11}
\end{aligned}
$$

Here $q_t(x)$ is the marginal density of $X_t$ computed from the approximate posterior path measure $q$. We have also defined the corresponding approximate *posterior drift*

$$g_t(x) = \lim_{\Delta t \to 0} \frac{1}{\Delta t} E_q[X_{t+\Delta t} - X_t | X_t = x], \tag{12}$$

as well as the functions

$$A(x) = \int_0^T q_t(x) dt \quad \text{and} \quad y(x) = \int_0^T g_t(x) q_t(x) dt. \tag{13}$$

There are two main problems for a practical realization of this EM algorithm:

1. We need to find tractable path measures $q$, which lead to good approximations for marginal densities and posterior drifts given *arbitrary* prior drift functions $f(x)$.

2. The M-Step requires a functional optimization, because (11) shows that $\mathcal{L}(\mathbf{f}, q) - \ln P_0(\mathbf{f})$ is actually a functional of $f(x)$, i.e. it contains a continuum of values $f(x)$, where $x \in \mathcal{R}^d$.

## 4.2 Linear drift approximation: The Ornstein-Uhlenbeck bridge

For given drift $f(\cdot)$ and times $t \in I_k$ in the interval $I_k = [k\tau; (k+1)\tau]$ between two consecutive observations, the exact posterior marginal $p_t(x)$ equals the density of $X_t = x$ *conditioned on* the fact that $X_{k\tau} = y_k$ and $X_{(k+1)\tau} = y_{k+1}$. This can be expressed by the transition densities of the homogeneous Markov diffusion process with drift $f(x)$. We denote this quantity by $p_s(X_{t+s}|X_t)$ being the density of the random variable $X_{t+s}$ at time $t + s$ conditioned on $X_t$ at time $t$. Using the Markov property, this yields the representation

$$p_t(x) \propto p_{(k+1)\tau - t}(y_{k+1}|x) p_{t-k\tau}(x|y_k) \text{ for } t \in I_k. \tag{14}$$

As functions of $t$ and $x$, the second factor fulfills a forward Fokker-Planck equation and the first one a Kolmogorov backward equation [13]. Both are partial differential equations. Since exact computations are not feasible for general drift functions, we *approximate* the transition density $p_s(x|x_k)$ in each interval $I_k$ by that of a process, where the drift $f(x)$ is replaced by its local linearization

$$f(x) \approx f_{ou}(x, t) = f(x_k) - \Gamma_k(x - x_k) \quad \text{with} \quad \Gamma_k = -\nabla f(x_k). \tag{15}$$

This is equivalent to assuming that for $t \in I_k$ the dynamics is approximated by the homogeneous *Ornstein-Uhlenbeck process* [13]

$$dX_t = [f(y_k) - \Gamma_k(X_t - y_k)]dt + D^{1/2}dW, \tag{16}$$

which is also used to build computationally efficient hierarchical models [14, 15], as in this case the marginal posterior can be calculated analytically. Here the transition density is a multivariate Gaussian

$$q_s^{(k)}(x|y) = \mathcal{N}\left(x|\alpha_k + e^{-\Gamma_k s}(y - \alpha_k); S_s\right) \tag{17}$$

where $\alpha_k = y_k + \Gamma_k^{-1}f(y_k)$ is the stationary mean and the variance $S_s = A_s B_s^{-1}$ is calculated using the matrix exponential

$$\begin{bmatrix} A_s \\ B_s \end{bmatrix} = \exp\left(\begin{bmatrix} \Gamma_k & D \\ 0 & -\Gamma_k^\top \end{bmatrix} s\right) \begin{bmatrix} 0 \\ \mathbf{I} \end{bmatrix}. \tag{18}$$

Then we obtain the Gaussian approximation $q_t^{(k)}(x) = \mathcal{N}(x|m(t); C(t))$ of the marginal posterior for $t \in I_k$ by multiplying the two transition densities, where

$$C(t) = \left(e^{-\Gamma_k^\top(t_{k+1}-t)}S_{t_{k+1}-t}^{-1}e^{-\Gamma_k(t_{k+1}-t)} + S_{t-t_k}^{-1}\right)^{-1} \quad \text{and}$$

$$m(t) = C(t)e^{-\Gamma_k^\top(t_{k+1}-t)}S_{t_{k+1}-t}^{-1}\left(y_{k+1} - \alpha_k + e^{-\Gamma_k(t_{k+1}-t)}\alpha_k\right)$$
$$+ C(t)S_{t-t_k}^{-1}\left(\alpha_k + e^{-\Gamma_k(t-t_k)}(y_k - \alpha_k)\right).$$

By inspecting mean and variance we see that the distribution is a equivalent to a bridge between the points $X = y_k$ and $X = y_{k+1}$ and collapses to point masses at these points.

Within this approximation, we can estimate parameters such as the diffusion $D$ using the approximate evidence

$$p(\mathbf{y}|\mathbf{f}) \approx p_{ou}(\mathbf{y}) = p(x_1) \prod_{j=1}^{n-1} q_\tau^{(k)}(y_{k+1}|y_k) \tag{19}$$

Finally, in this approximation we obtain for the posterior drift

$$g_t(x) = \lim_{\Delta t \to 0} \frac{1}{\Delta t} E\left[X_{t+\Delta t} - X_t | X_t = x, X_\tau = y_{k+1}\right]$$
$$= f(y_k) - \Gamma_k(x - y_k) + De^{-\Gamma_k^\top(t_{k+1}-t)}S_{t_{k+1}-t}^{-1}(y_{k+1} - \alpha_k - e^{-\Gamma_k(t_{k+1}-t)}(x - \alpha_k))$$

as shown in appendix A in the supplementary material.

### 4.3 Sparse M-Step approximation

To cope with the functional optimization, we resort to a sparse approximation for replacing the *infinite set* $\mathbf{f}$ by a sparse set $\mathbf{f}_s$. Here the GP posteriors (for each component of the drift) is replaced by one that is closest in the KL sense. Following appendix B in the supplementary material, we find that in the sparse approximation the likelihood (11) is replaced by

$$\mathcal{L}_s(\mathbf{f}, q) = \frac{1}{2}\int \|\mathrm{E}_0[f(x)|\mathbf{f}_s]\|^2 A(x) \, dx - \int \langle \mathrm{E}_0[f(x)|\mathbf{f}_s], y(x)\rangle \, dx, \tag{20}$$

where the conditional expectation is over the GP prior. In order to avoid cluttered notation, it should be noted that in the following results for a component $f^j$, the quantities $\mathbf{\Lambda}_s, \mathbf{f}_s, \mathbf{k}_s, \mathbf{K}_s^{-1}, y(x), \sigma^2$, similar to (7) depend on the component $j$, but not $A(x)$.

This is easily computed as

$$\mathrm{E}_0[f(x)|\mathbf{f}_s] = \mathbf{k}_s^\top(x)\mathbf{K}_s^{-1}\mathbf{f}_s. \tag{21}$$

Hence

$$\mathcal{L}_s(\mathbf{f}, q) = \frac{1}{2}\mathbf{f}_s^\top \mathbf{\Lambda}_s \mathbf{f}^s - \mathbf{f}_s^\top \mathbf{d}_s \tag{22}$$

with

$$\mathbf{\Lambda}_s = \frac{1}{\sigma^2}\mathbf{K}_s^{-1}\left\{\int \mathbf{k}_s(x) A(x) \mathbf{k}_s^\top(x)dx\right\}\mathbf{K}_s^{-1}, \qquad \mathbf{d}_s = \frac{1}{\sigma^2}\mathbf{K}_s^{-1}\int \mathbf{k}_s(x) y(x) \, dx. \tag{23}$$

With these results, the approximate *MAP* estimate is

$$\bar{f}_s(x) = \mathbf{k}_s^\top(x)(\mathbf{I} + \mathbf{\Lambda}_s \mathbf{K}_s)^{-1} \mathbf{d}_s. \tag{24}$$

The integrals over $x$ in (23) can be computed analytically for many kernels of interest such as polynomial and RBF ones. However, we have done this for 1-dimensional models only. For higher dimensions, we found it more efficient to treat both the time integration in (13) and the $x$ integrals by sampling, where time points $t$ are drawn uniformly at random and $x$ points from the multivariate Gaussian $q_t(x)$.

A related expression for the variance $\sigma_s^2(x) = K(x,x) - \mathbf{k}_s^\top(x)(\mathbf{I} + \mathbf{\Lambda} \mathbf{K}^s)^{-1} \mathbf{\Lambda}_s \mathbf{k}_s(x)$ can only be viewed as a crude estimate, because it does not include the impact of the GP fluctuations on the path probabilities.

## 5  A crude estimate of an approximation error

Unfortunately, there is no guarantee that this approximation to the EM algorithm will always increase the *exact* likelihood $p(\mathbf{y}|\mathbf{f})$. Here, we will develop a crude estimate how $p(\mathbf{y}|\mathbf{f})$ differs from the the Ornstein-Uhlenbeck approximation (19) to lowest order in the difference $\delta f(X_t, t) \doteq f(X_t) - f_{\mathrm{ou}}(X_t, t)$ between drift function and its approximation.

Our estimate is based on the exact expression

$$p(\mathbf{y}|\mathbf{f}) = \int dp_0(X_{0:T})\, e^{\ln L(X_{0:T}|\mathbf{f})} \prod_{k=1}^n \delta(y_k - X_{k\tau}) \tag{25}$$

where the Wiener measure $p_0$ is defined in (5) and the likelihood $L(X_{0:T}|\mathbf{f})$ in (6). The Ornstein-Uhlenbeck approximation (19) can expressed in a similar way: we just have to replace $L(X_{0:T}|\mathbf{f})$ by a functional $L_{\mathrm{ou}}(X_{0:T}|\mathbf{f})$ which in turn is obtained by replacing $f(X_t)$ with the linearized drift $f_{\mathrm{ou}}(X_t, t)$ in (6). The difference in free energies (negative log evidences) can be expressed exactly by an expectation over the posterior OU processes and then expanded (similar to a cumulant expansion) in a Taylor series in $\Delta \mathcal{L} = -\ln(L/L_{\mathrm{ou}})$. The first two terms are given by

$$\Delta \mathcal{F} \doteq -\{\ln p(\mathbf{y}|\mathbf{f}) - \ln p_{\mathrm{ou}}(\mathbf{y})\} = -\ln E_q\left[e^{-\Delta \mathcal{L}}\right] \approx E_q\left[\Delta \mathcal{L}\right] - \frac{1}{2}\mathrm{Var}_q\left[\Delta \mathcal{L}\right] \pm \ldots \tag{26}$$

The computation of the first term is similar to (11) and requires only the marginal $q_t$ and the posterior $g_t$. The second term contains the posterior variance and requires two-time covariances of the OU process. We concentrate on the first term which we further expand in the difference $\delta f(X_t, t)$. This yields

$$\Delta \mathcal{F} \approx E_q\left[\Delta \mathcal{L}\right] \approx \int_0^T E_q\left[\langle \delta f(X_t, t), f_{ou}(X_t, t) - g_t(X_t)\rangle\right] dt. \tag{27}$$

This expression could be evaluated in order to estimate the influence of nonlinear parts of the drift on the approximation error.

## 6  Experiments

In all experiments, we used different versions of the following general kernel, which is a linear combination of a RBF and a polynomial kernel,

$$K(x_1, x_2) = c\,\sigma_{\mathrm{RBF}} \exp\left(-\frac{(x_1 - x_2)^T (x_1 - x_2)}{2l_{\mathrm{RBF}}^2}\right) + (1 - c)\left(1 + x_1^\top x_2\right)^p, \tag{28}$$

where the hyperparameters $\sigma_{\mathrm{RBF}}$ and $l_{\mathrm{RBF}}$ denote the variance and length scale of the RBF kernel and $p$ denotes the order of the polynomial kernel.

Also, we determined the sparse points for the GP algorithm in each case by first constructing a histogram over the observations and then selecting the set of histogram midpoints of each histogram bin which contained at least a certain number $b_{min}$ of observations. In our experiments, we chose $b_{min} = 5$.

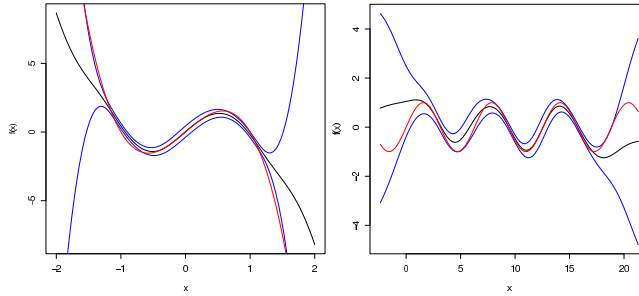

Figure 2: The figures show the estimated drift functions for the double well model (left) and the periodic diffusion model (right) after completion of the EM algorithm. Again, the black and blue lines denote mean and 95%-confidence bounds, while the red lines indicate the true drift functions.

## 6.1 One-dimensional toy models

First we test our algorithm on two toy data sets, the double well model with dynamics given by the SDE

$$dx = 4(x - x^3)dt + dW \qquad (29)$$

and a diffusion model driven by a periodic drift

$$dx = \sin(x)dt + dW. \qquad (30)$$

For both models, we simulated a path of size $M = 10^5$ on a regular grid with width $\Delta_t = 0.01$ from the corresponding SDE and kept every 20th sample point as observation, resulting in $N = 5000$ data points. We initialized the EM Algorithm by running the sparse GP for the observation points without any imputation and subsequently computed the expectation operators by analytically evaluating the expressions on the same time grid as the simulated path and summing over the time steps. An alternative initialization strategy which consists of generating a full trajectory of the same size as the original path using Brownian bridge sampling between observations did not bring any noticeable performance improvements. Since we cannot guarantee that the likelihood increases in every iteration due to the approximation in the E-step, we resort to a simple heuristic by assuming convergence once $\mathcal{L}$ stabilizes up to some minor fluctuation. In our experiments convergence was typically attained after a few ($< 10$) iterations. For the double well model we used an equal weighting $c = 0.5$ between kernels with hyperparameters $\sigma_{\mathrm{RBF}} = 1$, $l_{\mathrm{RBF}} = 0.5$ and $p = 5$, whereas for the periodic model we used an RBF kernel ($c = 1$) with the same values for $\sigma_{\mathrm{RBF}}$ and $l_{\mathrm{RBF}}$.

## 6.2 Application to a real data set

As an example of a real world data set, we used the NGRIP ice core data (provided by Niels-Bohr institute in Copenhagen, `http://www.iceandclimate.nbi.ku.dk/data/`), which provides an undisturbed ice core record containing climatic information stretching back into the last glacial. Specifically, this data set as shown in figure 3 contains 4918 observations of oxygen isotope concentration $\delta^{18}O$ over a time period from the present to roughly $1.23 \cdot 10^5$ years into the past. Since there are generally less isotopes in ice formed under cold conditions, the isotope concentration can be regarded as an indicator of past temperatures.

Recent research [16] suggest to model the rapid paleoclimatic changes exhibited in the data set by a simple dynamical system with polynomial drift function of order $p = 3$ as canonical model which allows for bistability. This corresponds to a meta stable state at higher temperatures close to marginal stability and a stable state at low values, which is consistent with other research on this data set, linking a stable state of oxygen isotopes to a baseline temperature and a region at higher values corresponding to the occurrence of rapid temperature spikes. For this particular problem we first tried to determine the diffusion constant $\sigma$ of the data. Therefore we estimated the likelihood of the data set for 40 fixed values of $\sigma$ in an interval from 0.3 to 11.5 by running the EM algorithm with a polynomial kernel ($c = 0$) of order $p = 3$ for each value in turn. The resulting drift function with the highest likelihood is shown in figure 3. The result seems to confirm the existence of a metastable state of oxygen isotope concentration and a stable state at lower values.

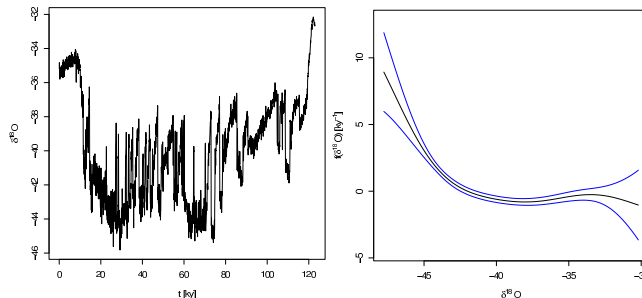

Figure 3: The figure on the left displays the NGRIP data set, while the picture on the right shows the estimated drift in black with corresponding 95%-confidence bounds denoting twice the standard deviation in blue for the optimal diffusion value $\hat{\sigma} = 2.9$.

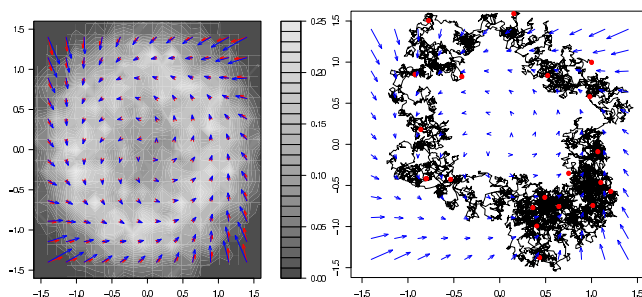

Figure 4: The left figure shows the empirical density for the two-dimensional model, together with the vector fields of the actual drift function given in blue and the estimated drift given in red. The right picture shows a snippet from the full sample in black together with the first 20 observations denoted by red dots.

### 6.3 Two-dimensional toy model

As an example of a two dimensional system, we simulated from a process with the following SDE:

$$
\begin{aligned}
dx &= (x(1 - x^2 - y^2) - y)dt + dW_1, \quad &(31)\\
dy &= (y(1 - x^2 - y^2) + y)dt + dW_2. \quad &(32)
\end{aligned}
$$

For this model we simulated a path of size $M = 10^6$ on a regular grid with width $\Delta_t = 0.002$ from the corresponding SDE and kept every 100th sample point as observation, resulting in $N = 10^4$ data points. In the inference shown in figure 4 we used a polynomial kernel ($c = 0$) of order $p = 4$.

## 7   Discussion

It would be interesting to replace the ad hoc local linear approximation of the posterior drift by a more flexible time dependent Gaussian model. This could be optimized in a variational EM approximation by minimizing a free energy in the E-step, which contains the Kullback-Leibler divergence between the linear and true processes. Such a method could be extended to noisy observations and the case, where some components of the state vector are not observed. Finally, this method could be turned into a variational Bayesian approximation, where one optimizes posteriors over both drifts and over state paths. The path probabilities are then influenced by the uncertainties in the drift estimation, which would lead to more realistic predictions of error bars.

**Acknowledgments**   This work was supported by the European Community's Seventh Framework Programme (FP7, 2007-2013) under the grant agreement 270327 (CompLACS).

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
