[Supplementary Material]

# Approximate Gaussian process inference for the drift of stochastic differential equations
# Supplementary material

**Andreas Ruttor**
Computer Science, TU Berlin
andreas.ruttor@tu-berlin.de

**Philipp Batz**
Computer Science, TU Berlin
philipp.batz@tu-berlin.de

**Manfred Opper**
Computer Science, TU Berlin
manfred.opper@tu-berlin.de

## A  Posterior drift

$$
\begin{aligned}
g_t(x) &= \lim_{\Delta t \to 0} \frac{1}{\Delta t} E\left[X_{t+\Delta t} - X_t | X_t = x, X_\tau = y\right] & (1)\\[2mm]
&= \lim_{\Delta t \to 0} \frac{1}{\Delta t} \frac{\int (x' - x)\, p_{\tau-t-\Delta t}(y|x') p_{\Delta t}(x'|x)\, dx'}{\int p_{\tau-t-\Delta t}(y|x') p_{\Delta t}(x'|x)\, dx'} & (2)\\[2mm]
&= \lim_{\Delta t \to 0} \frac{1}{\Delta t} \frac{f(x)\Delta t + E_u\left[p_{\tau-t-\Delta t}(y|x + f(x)\Delta t + u)u\right]}{E_u\left[p_{\tau-t-\Delta t}(y|x + f(x)\Delta t + u)\right]} & (3)\\[2mm]
&= f(x) + D \lim_{\Delta t \to 0} \frac{\nabla_x E_u\left[p_{\tau-t-\Delta t}(y|x + f(x)\Delta t + u)\right]}{E_u\left[p_{\tau-t-\Delta t}(y|x + f(x)\Delta t + u)\right]} & (4)\\[2mm]
&= f(x) + D \lim_{\Delta t \to 0} \nabla_x \ln\left\{E_u\left[p_{\tau-t-\Delta t}(y|x + f(x)\Delta t + u)\right]\right\} & (5)\\[2mm]
&= f(x) + D\nabla_x \ln\left\{p_{\tau-t}(y|x)\right\}. & (6)
\end{aligned}
$$

The second line follows from the definition of the conditional density, the 3rd line from the fact that $p_{\Delta t}(x'|x) = \mathcal{N}(x + f(x)\Delta t; D\Delta t)$ and $u \sim \mathcal{N}(0; \sigma^2 \Delta t)$. The fourth line is based on the fact that for zero mean Gaussian random vectors with covariance $S$, we have $E[ug(u)] = SE_u[\nabla_u g(u)]$. Finally, the last line is obtained by noting that as $\Delta t \to 0$, the covariance of $u$ vanishes.

## B  Kullback–Leibler optimal sparsity

### B.1  The general case

We assume a collection of random variables $\mathbf{f} = \{f(x)\}_{x \in T}$ where the index variable $x \in T$ takes values in some index set $T$. We will assume a *prior measure* denoted by $P_0(\mathbf{f})$ and a *posterior measure* of the form

$$
P(\mathbf{f}) = \frac{1}{Z} P_0(\mathbf{f})\, e^{-U(\mathbf{f})} \tag{7}
$$

where $U(\mathbf{f})$ is a functional of $\mathbf{f}$. The goal is to approximate $P$ by another measure $Q$ of the form

$$
Q(\mathbf{f}) = P_0(\mathbf{f})\, R(\mathbf{f}_s) \tag{8}
$$

where the effective likelihood $R$ depends only on a smaller, the *sparse* set $\mathbf{f}_s = \{f(x)\}_{x \in S}$ of dimension $m$. $S$ is not necessarily a subset of $T$. $R$ will be chosen to minimize the Kullback–Leibler divergence

$$
KL(Q||P) = \mathrm{E}_Q[\log(Q/P)]. \tag{9}
$$

We write the joint measure of $\mathbf{f}$ and $\mathbf{f}^S$ as

$$Q(\mathbf{f}, \mathbf{f}_s) = Q(\mathbf{f}|\mathbf{f}_s)Q(\mathbf{f}_s) = P_0(\mathbf{f}|\mathbf{f}_s)Q(\mathbf{f}_s), \tag{10}$$

where the last equality follows from the fact that fixing the sparse set $\mathbf{f}_s$, $R(\mathbf{f}_s)$ becomes nonrandom and the dependency on the random variables $\mathbf{f}$ is only via $P_0$. Hence, the KL divergence is obtained

$$KL(Q||P) = \ln Z + \int d\mathbf{f}_s Q(\mathbf{f}_s) \; \log \left( \frac{e^{\ln R(\mathbf{f}_s)}}{e^{-\mathrm{E}_0[U(\mathbf{f}|\mathbf{f}_s)]}} \right) \tag{11}$$

by integrating out all variables except $\mathbf{f}_s$. $\mathrm{E}_0[U(\mathbf{f}|\mathbf{f}_s)]$ is the conditional expectation w.r.t. the prior $P_0$. Hence, the optimal choice for $R$ is

$$R(\mathbf{f}_s) \propto e^{-\mathrm{E}_0[U(\mathbf{f}|\mathbf{f}_s)]} . \tag{12}$$

## B.2 Gaussian random variables

If $P_0$ is Gaussian measure and

$$U(\mathbf{f}) = \frac{1}{2}\mathbf{f}^\top \mathbf{\Lambda} \mathbf{f} - \mathbf{a}^\top \mathbf{f} \tag{13}$$

is a quadratic form, the posterior is also Gaussian. We can then further simplify the conditional expectation (12) to

$$\mathrm{E}_0[U(\mathbf{f})|\mathbf{f}_s] = \frac{1}{2}(\mathrm{E}_0\{\mathbf{f}|\mathbf{f}_s\})^\top \mathbf{\Lambda} \mathrm{E}_0\{\mathbf{f}|\mathbf{f}_s\} - \mathbf{a}^\top \mathrm{E}_0\{\mathbf{f}|\mathbf{f}^s\} + C \tag{14}$$

where $C = \frac{1}{2}\mathrm{tr}\left(\mathrm{Cov}_0\{\mathbf{f}|\mathbf{f}_s\}\mathbf{\Lambda}\right)$ is a constant independent of $\mathbf{f}_s$. This follows from the fact that for a Gaussian measures, all joint and conditional distributions are Gaussian, $\mathrm{E}_0\{\mathbf{f}|\mathbf{f}_s\}$ is the optimal mean square predictors of the Gaussian vector $\mathbf{f}$ given $\mathbf{f}_s$ [1]: and the difference $\mathbf{f} - \mathrm{E}_0\{\mathbf{f}|\mathbf{f}_s\}$ is a random vector which is *independent* of the vector $\mathbf{f}_s$. Hence the conditional covariance $\mathrm{Cov}_0$ of $\mathbf{f}$ does not depend on $\mathbf{f}_s$. The explicit result for this predictor is given by

$$\mathrm{E}_0[\mathbf{f}|\mathbf{f}^s] = \boldsymbol{\pi}\mathbf{f}_s, \tag{15}$$

where $\boldsymbol{\pi} = \mathbf{K}_{Ns}\mathbf{K}_s^{-1}$, $\mathbf{K}_s$ is the kernel matrix for the sparse set, and $\mathbf{K}_{Ns}$ is the $N \times m$ kernel matrix between the non-sparse and the sparse set.

For the infinite dimensional case of the form

$$U(\mathbf{f}) = \frac{1}{2}\int f^2(x)\Lambda(x)dx - \int f(x)y(x)dx \tag{16}$$

we use the fact that

$$\mathrm{E}_0[f(x)|\mathbf{f}_s] = \mathbf{k}_s^\top(x)(\mathbf{K}_s)^{-1}\mathbf{f}_s, \tag{17}$$

so that

$$\mathrm{E}_0[U(\mathbf{f})|\mathbf{f}_s] = \frac{1}{2}\mathbf{f}_s^\top \mathbf{K}_s^{-1} \left\{ \int \mathbf{k}_s(x)\,\Lambda(x)\,\mathbf{k}_s^\top(x)dx \right\} \mathbf{K}_s^{-1}\mathbf{f} - \mathbf{f}^\top \mathbf{K}_s^{-1} \int \mathbf{k}_s(x)\,a(x)\,dx. \tag{18}$$

## References

[1] Athanasios Papoulis. *Probability, random variables, and stochastic processes*. 1965.