[Reviews · NeurIPS 2013]

Submitted by Assigned_Reviewer_4

The Gaussian process is employed to infer the drift function in systems of stochastic differential equations (essentially Brownian motion, with non-constant drift), which are approximated in discrete form. The basic setup of the model is fairly straightforward, with the observed data assumed to be characterized by a Brownian motion, with the drift term a function of the state vector. It is assumed that a separate GP is appropriate for each dimension of the drift function, which may be a serious simplification. Sparse sampling is used to make GP inference tractable. Various approximations are needed to address the fact that the temporal sampling may not be fine enough. The main contribution appears to be use of EM to make the computations more efficient than sampling based methods.
Summary: The problem being considered is interesting, although it appears that the paper was completed in a bit of a rush. There were a few typos sprinkled throughout. More seriously, on p. 3 a figure label is "????". Also, technically speaking the paper is over-length, as the 9th page is supposed to be only reserved for references, and text spills over from p. 8. I am not overly worried about that, because there is a lot of wasted space in the size and usage of figures.

While you start out with a continuous time motivation and SDEs, once you get to (2) everything is discrete and one may just start there. Also, (4) and (5) come out directly from (2), and it may have been easier to explicitly write (3) as a multivariate Gaussian with mean f(X_t)\Delta t and covariance D\Delta t, which would allow the reader to immediately see that this is the standard Brownian motion model. As is, the discussion is a bit muddled, although not incorrect. The main contribution concerns modeling and inferring f(X_t), which you do in a very clear and (may I say) obvious way via independent GPs.

The experiments are nice, as are the bounds you put on performance.

Submitted by Assigned_Reviewer_6

The paper presents an interesting method for learning nonparametrically the drift of a stochastic differential equation model driven by Brownian noise. This is based on a recent observation that for a fully observed diffusion path, the so called likelihood of the SDE has a quadratic from wrt the drift function. Then one can assume a GP prior over the drift and turn this problem into GP regression estimation. However, having a fully observed diffusion path
is unrealistic (typically only a finite set of observations are given) and the authors propose an EM algorithm for the estimation of the drift where diffusion sub-paths between consecutive observations are integrated out in the E-step.

The paper is mainly clearly written and the authors present an interesting method to solve a very challenging problem. Some comments are given below.

As the authors point out, the EM algorithm does not quaranteed to improve the likelihood in each step, because of the OU linearization needed to carry out the E-step. Also it is unclear why the authors have preferred to approximate equation 24 by using Monte Carlo, despite the fact that this integral is analytically tractable for the squared-exponential kernel.

The writing in the experiments section contains significantly many more typos and spelling mistakes than the rest of the papers. It seems that it has been written in a hurry. Despite that the experiments are interesting and the method has the ability, for instance, to learn reasonably well the sine function in Figure 2.

Summary: An EM algorithm for learning the drift function in SDEs. Some bits of the approximations are questionable, but the results seems to be promising.

Submitted by Assigned_Reviewer_7

This is a solid paper on a topic of potential significant interest. Diffusion processes are widely used in the sciences and engineering, and a data-driven strategy to parametrize the drift function could be potentially useful. Furthermore, the paper also has considerable interest methodologically: the proposal to approximate the intractable problem with a piecewise OU process is interesting, and the approximate quantification of the error is (at least in theory) a nice result. I would say the paper scores highly on quality and originality. It is generally clearly written although some bits feel a bit rushed (see detailed comments below). Re significance, I would say the methodology could be of considerable interest, while the experiments are for the time being a bit less well developed.

Detailed comments:
- I would have appreciated a bit more motivation for the methodology: SDEs in the sciences are often used as a modelling tool so that the specific form of the drift has a physically interpretable meaning. This is somewhat lost with the GP approximation;
- I wonder whether there are some asymptotic results which could be given, at every step you effectively get a measurement of the linearised drift (at a different input value), hence of its value and its derivative. When you increase the number of observations, this should converge to the true posterior (which of course it does as the approach becomes exact for observed paths), but perhaps you can work out the rate at which your approach converges to the truth?
- The experimental part is a bit disappointing, there are essentially no comments on the performance of the model on simulated data, and very little context on the real data. I would remove the experiment with 5th degree drift and leave more space to comment on the experiments. Also some runtimes would be useful.
Minor comments:
- eq(10) worth perhaps saying that the last step uses explicitly the time invariance of the drift function (homogeneous GP);
- after (13), it's the two terms on the r.h.s. that obey Ch-K equations (the way it's written is ambiguous);
- some confusion with figures, there is a reference to Figure ?? (1a presumably) on page 3; the caption to Figure 2 alludes to a missing third panel;
- models that use OU processes with piecewise constant drifts have been considered and shown to be very computationally efficient (Stimberg et al AISTATS11, Opper et al NIPS10 which reported very fast performance on the double well problem). These are different as the changepoints are stochastic processes while here the changes are an approximation strategy, but they seem somewhat related and might be worth referring to.
Summary: A solid paper with nice methodological developments in an interesting area. In my opinion, the originality compensates for the somewhat limited experimental part.
Author Feedback

Author rebuttal: We thank all reviewers for their comments. We are going to fix the layout issues, the missing label on page 3, and the typos in the final version.

The figure referenced by the label "figure ??" on page 3 in line 134 is the right plot of figure 1 on page 7. It shows the difference between modelling a transition kernel (red line) for discrete time steps and the drift function (black line) of a continuous time process. Therefore it is important to use equation (2) only in the limit \Delta t --> 0. As this limit can be calculated analytically, \Delta t is not needed in the final equations for estimating the drift function. This is an advantage, as step sizes for numerical calculations are independent of the model.

We will address the specific concerns of each reviewer in turn.

@Reviewer 4:

We are sorry that we created the impression that in our work stochastic differential equations "are approximated in discrete form". This is not the case! The Euler discretization equation (2) was introduced to motivate the likelihood of the data avoiding more technical approaches of stochastic analysis such as Girsanov's change of measure theorem. In our main results, equations (10) and (11) and the approximating OU bridge, the continuous time limit \Delta t \to 0 was explicitly assumed.

In order to clarify the relation between equations (2)-(5), we are going to specify the transition probabilities explicitly in equation (3). The split into prior (4) and likelihood (5) is shown there in order to use it later in the description of the EM algorithm.

We only discuss the case of independent GPs in the paper, as that reflects best the typical prior knowledge about the drift function: we can guess the range of input and output values and use that to specify prior variance and length scale of an RBF kernel. But we assume that the components of the drift function are independent.

Nevertheless, replacing the set of K^j with a kernel for all components is possible in our implementation, but increases the size of the matrices and therefore the computational complexity. Then the algorithm can also cope with non-diagonal diffusion matrices, which introduce correlations between the drift components in the posterior.

@Reviewer 6:

While the integrals given in equation (24) are analytically tractable for many kernels, especially RBF and polynomial ones as noted in line 286, doing this analytical integration is time-consuming and error-prone. Additionally, it may not work for a special kernel, which has been chosen due to prior knowledge about the behavior of the observed system. Therefore we show how one can do the computation in the general case by Monte-Carlo integration, which only requires a small number of independent samples until convergence is reached.

@Reviewer 7:

As mentioned in line 032 SDEs are a suitable model for dynamical systems. We focus on estimating the drift function, especially because it determines the main part of the dynamics and typically has a direct (physical) meaning. We are going to add this remark to the introduction.

Thank you for suggesting the computation of asymptotic rates. We expect that this should be possible by a combination of an expansion of the error (section 5) with respect to small \tau and the asymptotic results of Bayes estimators given in reference [9].

Stimberg et al (AISTATS 2011) and Opper et al (NIPS 2010) include the OU process with piecewise constant parameters directly in their model, while we only use it for the approximation. The main advantage in all cases is that this process is analytically tractable, which allows for fast computations. We are going to add these references as other uses of such an OU process.